# Cancer-Cell-Derived IgG and Its Potential Role in Tumor Development

**DOI:** 10.3390/ijms222111597

**Published:** 2021-10-27

**Authors:** Said Kdimati, Christina Susanne Mullins, Michael Linnebacher

**Affiliations:** Clinic of General Surgery, Molecular Oncology and Immunotherapy, University Medical Center Rostock, University of Rostock, 18057 Rostock, Germany; Said.Kdimati@med.uni-rostock.de (S.K.); christina.mullins@med.uni-rostock.de (C.S.M.)

**Keywords:** tumor-derived IgG, tumor marker, tumor neoantigens, cancer-cell-derived IgG

## Abstract

Human immunoglobulin G (IgG) is the primary component of the human serum antibody fraction, representing about 75% of the immunoglobulins and 10–20% of the total circulating plasma proteins. Generally, IgG sequences are highly conserved, yet the four subclasses, IgG1, IgG2, IgG3, and IgG4, differ in their physiological effector functions by binding to different IgG-Fc receptors (FcγR). Thus, despite a similarity of about 90% on the amino acid level, each subclass possesses a unique manner of antigen binding and immune complex formation. Triggering FcγR-expressing cells results in a wide range of responses, including phagocytosis, antibody-dependent cell-mediated cytotoxicity, and complement activation. Textbook knowledge implies that only B lymphocytes are capable of producing antibodies, which recognize specific antigenic structures derived from pathogens and infected endogenous or tumorigenic cells. Here, we review recent discoveries, including our own observations, about misplaced IgG expression in tumor cells. Various studies described the presence of IgG in tumor cells using immunohistology and established correlations between high antibody levels and promotion of cancer cell proliferation, invasion, and poor clinical prognosis for the respective tumor patients. Furthermore, blocking tumor-cell-derived IgG inhibited tumor cells. Tumor-cell-derived IgG might impede antigen-dependent cellular cytotoxicity by binding antigens while, at the same time, lacking the capacity for complement activation. These findings recommend tumor-cell-derived IgG as a potential therapeutic target. The observed uniqueness of Ig heavy chains expressed by tumor cells, using PCR with V(D)J rearrangement specific primers, suggests that this specific part of IgG may additionally play a role as a potential tumor marker and, thus, also qualify for the neoantigen category.

## 1. Introduction

Immunoglobulins (Ig), commonly known as antibodies, are, according to classical theory, proteins secreted by B lymphocytes (B cells), which, in turn, are dedicated to the adaptive immune system. Constituting 75% of all Igs and 10–20% of all circulating proteins, Igs of the class G are the most abundant proteins in human serum [1]. They are composed of 82–96% amino acids and 4–18% carbohydrate, thus closely resembling glycoproteins. Despite an over 90% identity at the amino acid level of the four IgG subclasses, IgG1, IgG2, IgG3, and IgG4, each one of the subclasses has its own unique antigen recognition and antigen binding profile [2]. Igs are disposed in a Y-shaped conformation and composited of four parts, two identical light chains (IgL) adjunct to two identical heavy chains (IgH). Each IgG arm carries variable regions, facilitating the recognition of a specific antigenic structure due to the assembly of the variable regions by the V(D)J recombination process. Therefore, different gene segments are rearranged and assembled to form the constant region, which, in turn, defines the Ig class [3]. Furthermore, the constant Fc part of an antibody forms a bond with specific receptors on other immune cells, thus executing the Ig’s effector function [4]. Currently, human polyvalent IgG is used for treatment of antibody deficiencies, including the B cell lymphoproliferative syndromes [5,6]. In addition, idiopathic thrombocytopenic purpura, Kawasaki disease, inflammatory demyelinating polyradiculoneuropathy, and childhood encephalitis are treated with IgG from donor pools [7,8,9,10]. According to this, IgG is not solely used in immune impaired patients, but is also implemented in the treatment of neurological disorders. Both intravenous and subcutaneous administration of IgG were established decades ago and are still in use [11].

Ig synthesis of the five known classes (IgA, IgM, IgG, IgD, and IgE) is thought to be realized solely by B cells. However, Igs of the classes IgG, IgA, and IgM, as well as their corresponding mRNAs, could be detected in both the cytoplasm and the supernatant of malignant non-B cells [12,13,14,15,16,17,18]. Additionally, in a few cases, the expression of whole IgG, or at least entire Ig chains, was demonstrated also in noncancerous, nonlymphoid tissues, such as normal human lung and colon tissues [12,14,19]. The functions of these non-lymphoid and noncancerous Igs are still unclear. Similarly, the detailed role of the cancer-derived Igs (CIg) is still largely unexplored; there is, however, evidence of a cancer-promoting effect. This effect becomes apparent through the observed increase of tumor cell apoptosis, as well as the decrease of colony formation and invasion after intermitting the production of cancer-derived IgG (CIgG) using anti-human IgG antibodies and siRNA [12,17]. The notable difference between Igs produced by B cells and CIgs produced by cancer cells manifested as restricted patterns of V(D)J recombination in CIgGs and Igs of normal tissue controls compared to IgGs originating from B cells [14]. Distinguishing B-cell-derived IgG from CIgG is possible with RP215, an antibody that specifically detects CIgG and shows a strong anticancer effect in vivo [20,21]. The uniqueness of the expressed Ig repertoire opens the opportunity to use tumor-specific V(D)J patterns not only as biomarkers, but also as (immune-)therapy targets due to their neoantigenic characteristics. In addition to directly solid tumor-derived Igs, an especially high burden of malignant plasma cells can be the origin of extraordinary Igs. Plasma cell neoplasms lead to a high number of abnormal plasma cells, either benign or malignant. Common neoplasms are the monoclonal gammopathy of undetermined significance (MGUS) and multiple myeloma [22]. MGUS-derived B cells secrete large amounts of monoclonal Igs. Multiple myeloma is known to be derived from MGUS and possesses a 1% expression rate of some Ig types [23,24,25]. The products of these malignant B cells are mostly IgG, followed by IgA, solitary free light chains (Bence Jones myeloma), and IgD; all detectable in serum or urine samples [26]. The abnormal amount of Igs causes renal insufficiency and amyloidosis, accompanied by proteinuria [27]. The relative decrease of polyclonal antibodies leads to an immunodeficiency of the patients [28].

## 2. Current State of Knowledge

### 2.1. CIgs Use a Limited Repertoire

In general, CIgs show classical and functional V_H_DJ_H_ rearrangement patterns, comparable to B-cell-derived Igs. However, the diversity of IgVH known from conventional IgG is not achieved in CIgs, based on the higher hypermutation rate of B-cell-derived IgG [29]. When comparing the expression profile of the amplified IgH of five pairs of colon adenocarcinomas and corresponding normal samples, Ig expression rates of normal epithelial cells in descending order of IgA (46%), IgG (28.35%), IgM (13.77%), and IgD (11.86%) were observed. IgE was detected in three samples, with a mere mean proportion of 0.02%. In contrast, the cancerous cells mainly expressed IgG with a mean proportion of 87% [14]. When compared to the V_H_DJ_H_ patterns of conventional Ig produced by B cells [30], the IgH repertoire found in cancerous cells and corresponding normal cells was rather limited. Nevertheless, in every analyzed case, the cancer-cell-derived IgH differed from that of the matched normal epithelial cells. Genetic analyses showed a more frequent expression of V_H_ segments close to J_H_ segments in cancer cells, prompting the preferential use of V_H_ segments in cancer cells. These are the evolutionarily older segments. Further investigations revealed that no identical V_H_DJ_H_ patterns were shared by samples of different patients. In contrast, several patterns were identified in different normal cell samples of different patients. Additionally, the IgH sequences of cancer cells were significantly mutated in the V_H_3–23 region compared to normal cells [14].

### 2.2. Ig Repertoire in Human Cancers

Several research groups identified IgG expression in tumor tissues originating from different organs. In colorectal cancer (CRC) patients, all five classes of IgHs and the entire Ig repertoire were detected by multiplex PCR and immune repertoire sequencing. Additionally, in corresponding normal colon cell samples, IgH was also expressed [14]. Qiu et al. detected IgG in tissue derived from different malignant epithelial tumors (breast, liver, lung, and colon). In all tumor samples (*n* = 42), IgG was located in the cytoplasm but not in any of the normal tissues. IgG was detected in some basal layer cells in close proximity to tumor cells in only 4 out of 10 samples from normal lung tissue. To exclude IgG contamination originating from B cells, long-term cell cultures were also examined. Flow cytometry analyses revealed IgG expression in multiple long-term cultured human epithelial cancer cell lines (breast, colon, liver, ovarian, lung, and cervix). Additionally, the supernatants of two cervical cancer cell lines tested positive for IgG in ELISA [12]. In 68 breast cancer samples, the abundance of IgG-expressing tumor cells was higher in metastasized samples than in matched primary tumors [15]. Most recently, Jiang et al. confirmed these findings using the RP215 antibody; they detected CIgG in the cytosol, the cell membrane, and the extracellular matrix in five tumor cell lines. By using anti-human IgG antibodies, IgL could be detected in all five analyzed cell lines, while the heavy chain was found only in one [16]. From 150 CRC samples, 12 revealed expression of IgG light and heavy chains, whereas the corresponding normal tissues were negative for IgG. The Igs were primarily located in the cytoplasm. In mesenchymal connective tissue, except the smooth muscle, IgG was also detectable [17].

Chen and co-authors investigated the expression of IgG in a total of 80 tissue samples of soft tissue malignancies, including fibrous histiocytomas, malignant fibrous histiocytomas, fibromas, fibrosarcomas, leiomyomas, leiomyosarcomas, and rhabdomyosarcomas, as well as 10 samples of normal striated muscle. Antibodies against Igy and Igk were used to detect IgL expression. A significant difference was observed, with a mere 31.7% of normal and benign tumor tissues expressing Igk, compared to 97.4% of malignant tumors. Again, the Igk was found mainly to be located in the cytoplasm and on the cell membrane [31].

In addition, Qiu et al. detected IgL in the cytoplasm of papillary thyroid cancer tissue. Here, IgH was detected in 80% and IgL in 59% of samples, using a total of 44 samples. No IgG was detected in the corresponding normal tissues [18].

To investigate the amount of CIgG in the human circulation, over 500 serum samples were analyzed for a CIgG component (CA215) by Lee et al. in 2010. Patients from 8/12 different cancer types revealed significantly higher CA215 levels in serum than healthy controls (see Figure 1). The positivity rates were in descending order: lymphoma (83%), liver cancer (74%), breast cancer (71%), esophageal cancer (61%), stomach cancer (60%), ovarian cancer (59%), lung cancer (52%), cervical cancer (51%), colon cancer (44%), pancreatic cancer (41%), prostate cancer (40%), and kidney cancer (38%) [32]. Taken together, these results indicate a wide range of tumor entities with the ability to express CIgG. An overview, including additional results, is given in Table 1.

Recently, our group investigated the expression of IgG in 12 low-passage colon cancer cell lines established from 11 patients. Despite the fact that the addition of commercially available IgG did not influence the viability of any of the tested cell lines, anti-viability effects of oxaliplatin were significantly abrogated following incubation of the cell lines for more than 5 d with IgG. Additionally, a significantly reduced rate of cell death was measured with Annexin V/PI staining when cells were treated with IgG and oxaliplatin. Further investigations revealed that pooled normal human IgG and IgG1, which is included in the applied IgG formulation, significantly inhibited the anticancerous activity of oxaliplatin, too. In two cell lines with high IgG expression levels, a significant correlation of IgG expression and decrease of oxaliplatin sensitivity was detected [51].

### 2.3. Regulation of IgG Expression in Tumors

The regulation of Ig expression in B cells has been well investigated. Multiple transcription factors, such as E-box protein E2A, early B-cell factor (EBF), octamer-related protein-1 (Oct-1), and Oct-2, regulate the expression of the heavy and light chain loci. In 2010, Zhu et al. detected E2A expression in all investigated non-B-cell cancer cell lines. E2A activates B-cell gene expression and IgH gene rearrangement and, thus, drives the generation of the earliest B-cell progenitors. Oct-1, which can be found during early B-cell development and the development of other cells, was also identified as being widely expressed in cancer cell lines. EBF and Oct-2 were also detected in a few cell lines. Furthermore, Ig gene variable region promoters seem to primarily contain a motif termed the octamer element (5′-ATGCAAAT-3′). The location of this element has been described as 10–25 nucleotides upstream of the TATA box. It is essential in promoting Ig gene transcription in B cells and is conserved in all Ig heavy- and light-chain variable region promoters [52,53]. A point mutation in the region of this octamer element reduced the Ig transgene expression in genetically modified mice to a mere 5% of the normal level [53]. These results revealed that the octamer element for Oct-1 and Oct-2 binding is present in non-B-cell cancer cell lines and is mandatory for Ig gene expression not only in B cells but also in non-B-cell cancer cells [54]. When summing up these data, it seems plausible that CIgG expression is orchestrated by the normal regulatory pathways and factors, which are pathologically active and, thus, hijacked by tumor cells, similar to classical signaling pathways such as the EGFR pathway.

### 2.4. Correlation of CIgG Expression with Clinicopathological Characteristics

In 2012, Niu et al. compared the clinicopathological features of 150 CRC cases [17]. Generally, no significant correlation between IgG expression and age, gender, cancer location, or cancer growth pattern were observed. Similarly, there were no correlations between the IgG expression and the individual pTNM stage. However, poorly differentiated adenocarcinomas more frequently expressed high IgG levels than moderately and well-differentiated ones. By comparing cancers with weak versus strong inflammatory infiltration, the weak ones tended to have a higher IgG expression than the strongly infiltrated ones. CRC cases with lymph node stages N_1_ and N_2_ showed a significantly higher rate of strong IgG expression than N_0_ cases. In addition, the expression of IgG was positively correlated with the expression of Cyclin D1, NF-kB, and PCNA. It was negatively correlated with the expression of Bcl-2. There was no correlation observed with the expression of MMP-2 or the p53 proteins [17]. Another study observed strong IgG staining in 89% of high-grade versus 63% in low-grade bladder cancer cases. Larger tumors had a higher IgG expression than smaller ones (94% vs. 70%). Moderate to strong expression of IgG was also associated with cancer recurrence; 63% of recurrent cases stained positive [42]. In a tissue microarray immunohistochemistry study, CIgG was more frequently expressed in the cancer nests of the malignant tissues (55%) compared to the matching normal tissues (7.5%). Using Kaplan–Meier survival analysis, the correlation between CIgG expression levels and overall survival (OS) of patients had been investigated. The OS was negatively correlated with moderate and high CIgG expression levels compared to low expression levels. In multivariate analyses, CIgG staining intensity in CRC tumors, lymph node metastases, TNM stage, and tumor location were independent prognosticators of 5-year OS [16]. In 2012, Qiu et al. reported a higher expression of IgG heavy chain in cancer tissue compared to normal papillary tissue and described a correlation between positive staining, staining intensity, and cancer size. Additionally, cases with local lymph node metastases had a significantly stronger IgH staining intensity than cases without [18].

### 2.5. Effects of CIgG

So far, comparably few studies have analyzed functional implications of CIgG expression. The observed effects can be divided into two categories: (I) direct effects on tumor cells and (II) systemic effects. To determine the role of CIgG in cancer progression, the expression of CIgG heavy chains was downregulated in SW480 cells using siRNA. This significantly inhibited cell proliferation, colony formation capacity, wound healing, and invasion [16]. Antisense oligonucleotides complementary to FR3 of the Ig variable region heavy chain were used to inhibit IgG expression in A549, HT-29, and HeLa MR cells. The inhibition of CIgG expression ranged from 30% to 80%. Tumor cell apoptosis significantly increased in all siRNA-treated cell lines. In MTT assays, the incubation with goat anti-human IgG inhibited the proliferation rate of cancer cell lines. Treatment of HeLa MR xenograft tumors with goat anti-human IgG antibodies significantly delayed tumor growth. Extensive necrosis and apoptosis were observed in histological assessments [12]. These results suggest a more active role of CIgG in cancer cell survival.

In 2019, Jiang and colleagues treated SW480 cells with siRNAs targeting Ig γ chains using lentiviral vectors and verified the knockdown efficiency by immunoblot analysis. The treated SW480 cells proliferated, migrated, and invaded less compared to controls treated with empty vectors. When these cells were injected into nude mice, the average tumor size and weight was significantly lower in the knockdown group compared to the control group [16].

Similar results were achieved by Niu et al. when treating LoVo cells with siRNA complementary to the constant region of CIgG or with anti-IgG antibodies. Beside downregulated IgG expression, proliferation was suppressed and at the same time the proportion of apoptotic LoVo cells increased [17]. Furthermore, IgH knockdown significantly decreased proliferation, migration, and invasion abilities of two bladder cancer cell lines in comparison to controls [42]. Similarly, viability and colony formation capacity of HeLa cells significantly decreased via siRNA targeting of IgH [41]. In summary, CIgG seems to elicit autocrine and paracrine effects. The congruent in vitro effects of CIgG described by different groups are summarized and schematically presented in Figure 2.

Other groups showed that sialylated CIgGs (SIA-CIgG) were able to increase tumor volume over time in immunocompetent mice, but not in those that were immunodeficient, compared to control mice treated with PBS. T-cell numbers in the lymph nodes of the treated immunocompetent mice, but not the immunodeficient animals, were decreased compared to the controls. Most importantly, treatment of immunocompetent mice with SIA-CIgG in combination with RP215 restored the lymph node T-cell numbers and attenuated tumor growth. The same group injected murine melanoma cells into immunocompetent and immunodeficient mice and treated them with IgG, isolated from the tumor microenvironment (TME). This resulted in TME-IgG-promoted tumor growth and simultaneous reduction of CD4^+^ and CD8^+^ T cells in the lymph nodes of the immunocompetent mice. No significant differences were detectable in the immunodeficient mice [21]. All in vivo results are summed up and schematically shown in Figure 3.

The above-listed findings all revolve around effects on the tumor cells themselves. Concerning peritumoral or even systemic effects of CIgG, very little is known. Some data suggest a mode of action for CIgG on T cells via sialic-acid-binding immunoglobulin-type lectins (siglecs). These are considered to be more important in the direct tumor microenvironment than in the circulation. Additionally, direct effects of CIgG on platelets have been described. This hints towards not only direct, but also systemic effects. Several groups have investigated the effects of platelets on tumors. Here, platelets encase circulating tumor cells, thereby protecting them from NK-cell-mediated cytolysis [55].

For a stable adhesion, tumor cells can secrete platelet-activating substrates, like ADP, and proteins, like thromboxane A2 or high-mobility group box 1. These factors initiate activation via toll-like receptor 4 binding [55,56,57]. Platelets can additionally regulate tumor growth, angiogenesis, and metastasis [58,59,60]. They influence tumor cells due to their high number of surface receptors and secreted molecules; including thromboxane, platelet-derived growth factor, and vascular endothelial growth factor [61,62,63,64]. When activated, P-selectin (CD62P) moves from an internal location to the cell surface [65,66,67]. Blocking CD62P resulted in a decrease of metastatic colon cancer foci in the lungs of mice and a diminished binding of platelets to tumor cells, indicating that CD62P is directly involved in the protumoral properties of platelets [68,69,70]. Maio et al. showed that CD62P expression was significantly higher in patients than in healthy donor blood samples. Additionally, supernatants of different cancer cell lines (SiHa, EJ, and SMMC.7721) could activate platelets. When the cell lines were treated with siRNA to suppress IgG1 expression, platelet activation was reduced (see Figure 4). Besides, human recombinant IgG did not induce the expression of CD62P, thus confirming the tumor-specific effect of CIgG on platelet activation [71]. CD62P expression, induced by cancer cell culture supernatant, led to FcγRIIa activation via phosphorylation, which, in turn, was important for strong immune and inflammatory responses. Blocking FcγRIIa reduced CD62P expression induced by the supernatants. This was achieved by treatment with siIgG1 supernatants and with cancer cell lysates depleted of CIgG by immunoprecipitation with RP215. Both conditions failed to activate platelets and the inability of RP215 to bind to normal IgG strongly suggests that FcγRIIa might have directly interacted with CIgG [71]. These findings strengthen the hypothesis that tumors produce CIgG to activate platelets in order to facilitate a metastatic spread by protecting the tumor cells from physical shear stress, as well as the immune surveillance in terms of NK and possibly also T-cell-mediated tumor cell lysis.

### 2.6. IgG Signaling

#### 2.6.1. Physiological

In order to understand the effects of CIgG in more detail, it might be helpful to have a closer look at the signaling pathway of normal IgG. All Igs, especially IgGs, are capable of linking the innate immune system to the adaptive one by forming a bridge between antigen-binding sites of the antibodies with various innate receptor and adaptor molecules. Thereby, IgG effects the modulation of adaptive immune responses and enhances T-cell activation, as well as the selection of high-affinity B cells. The most important receptors for IgG are the FcγR molecules. For binding with these receptors, the region of amino acids of the N-terminus of the CH2 domains and the amino acids located adjacent in the three-dimensional fold are required [72,73]. Additionally, each of the IgG subclasses has its own unique binding profile to FcγRs [74]. Defects in this complex system typically result in a variety of autoimmune and inflammatory conditions, as well as in defects in host defense responses [75]. Type 1 FcγRs can be subdivided into activating and inhibiting receptors, depending on the presence of intracellular activating or inhibiting signaling motifs. FcγRI, FcγRIIa, and FcγRIIIa as activating and FcγRIIb as inhibiting receptors are known to be expressed in humans. Here, FcγRI has the highest affinity to IgG. Activation of these receptors regulates mechanisms such as antibody-dependent cellular cytotoxicity, phagocytosis, and proinflammatory cytokine production. As discussed above, FcγRIIa plays an important role in platelet activation [71], while FcγRIIb lowers inflammatory immune responses [76]. FcγRs of type 2 are C-type lectins and were identified as IgG receptors only recently [77,78]. Representatives of type 2 FcγRs are mouse orthologue SIGN-R1 (DC-SIGN) and CD23, which both have an oligomeric structure stabilized through an α-helical coiled-coil stalk motif at the extracellular receptor domain. For the binding of IgG to type 2 FcγRs, the presence of sialylated Fc glycoforms is essential for destabilizing the Fc structure, enabling a conformation change of the Fc domain and the ligation to type 2 receptors. Like most of the type 1 receptors, type 2 receptors show a low affinity to their Fc ligand. The immunomodulatory effects of type 2 receptors are mediated by cytokine production via DC-SIGN and high-affinity B-cell selection via CD23 [79,80].

The signaling pathway activated by the interaction of IgG and FcγRs depends on the receptor and the cell type engaged in the interaction. The affinity of the receptor for IgG is determined by the Fc domain of the antibody and the associated N-linked glycan [76]. All cytotoxic responses are mediated by type 1 FcγRs. The linkage between type 1 FcγRs and an IgG-opsonized target initiates a signaling cascade, which leads to the destruction of the target [81,82]. After stimulation, type 1 FcγRs are able to oligomerize and crosslink, which in turn leads to the sequestration of FcγR-IgG complexes in intracellular vesicles for lysosomal degradation and antigen processing [83]. Generally, all type 1 FcγRs, except the FcγRIIb1 splice variant, have the ability to initiate phagocytosis of IgG-opsonized targets. Thus, activating type 1 FcγRs can mediate more potent reactions [84,85,86,87]. This includes proinflammatory cytokine and chemokine production and modulation of leukocyte differentiation, activation, and survival. The uptake of IgG-coated targets is associated with enhanced endosomal maturation and lysosomal fusion. Consequently, the antigen processing and presentation on MHC class II molecules occurs in a more efficient way [85,88]. It is known that antigen processing and presentation, due to FcγR-mediated internalization of immune complexes, increases T-cell activation [89,90,91,92]. Summarized, IgG is able to augment T-cell activation through target opsonization and T-cell recognition by activating type 1 FcγRs. The known and described receptors for CIgG are collectively shown in Figure 5.

#### 2.6.2. CIgG

Intravenous immunoglobulin (IVIG) negatively regulates T-cell-mediated immune responses independent of FcγR binding. These immunomodulatory effects are likely caused by a small fraction of IgG, accounting for 1–5% of the IVIG. The anomaly of these specific IgG molecules is an additional sialic acid added to the Fc region of the antibodies [93,94]. Dendritic-cell- and macrophage-mediated T-cell activation has been reported to be negatively impacted by this IVIG fraction [95,96,97]. The reason for this for seems to be the expression of glycan binding proteins on the surface of immune effector cells, called siglecs. Sixteen sialic acid receptors are known to be expressed on monocytes, myeloid, dendritic, NK, B, and T cells and have been suggested to mediate the immunosuppressive effects of sialylated IgG [98,99]. Inhibitory siglecs are expressed on T cells, rendering them putative targets for sialylated IgG. Wang et al. demonstrated that SIA-CIgG suppresses the proliferation of activated CD4^+^ and CD8^+^ T cells in a dose-dependent manner in vitro. The suppression was weaker when the sialic acid residues of the SIA-CIgG were removed by neuraminidase. In vitro, the proliferation rate of B16 melanoma cells was not affected by SIA-CIgG, thereby implying that their protumoral activity must be attributed to their effects on T cells. In line with this, antibodies directed against inhibitory siglecs fully restored T-cell proliferation capacity in the presence of SIA-CIgG in vitro. These results, coupled with increased siglec expression in the blood of cancer patients, make siglecs possible targets for immunotherapy. Furthermore, siglecs act as potential contact points for cancer cells to initiate immune escape mechanisms [21].

### 2.7. RP215

For the specific detection of CIgG, it is necessary to clearly distinguish between IgGs truly synthesized by cancer cells and IgG molecules ordinarily synthesized by B cells. When investigating CIgG expression, many groups used antibodies against conventional human IgG. To ensure that there is no contamination by conventional IgG, predominantly as a result of tumor-infiltrating B cells, a co-staining or co-analysis of B cells is necessary. However, this is no 100% guarantee for omitting IgG derived from the circulation. Using classical hybridoma technology, several murine monoclonal antibodies against the ovarian cancer cell line OC-3-VGH were generated. The best candidate antibody with a high affinity and specificity for ovarian cancer cells and a low cross-reactivity with most normal human tissues is RP215. It has also been shown to specifically bind the tumor-associated antigen COX-1, which is expressed by certain ovarian and cervical cancer cell lines [100]. In detail, RP215 was shown to interact with carbohydrate-associated epitopes of cancer-cell-derived glycoproteins known as CA215, found in cancer-derived Igs, but not in normal Igs [101].

These CIgs contain significantly higher amounts of N-glycoyl neuraminic acid in the O-linked glycans, but lower amounts of N-acetylglucosamine in the N-linked glycans compared to normal IgG. Immunoblot, carbohydrate profiling, and enzyme immunoassays proved that CA215 could, similarly to the variable heavy chains of human Igs, be detected with molecular sizes ranging from 50 to 70 kDa by RP215. Results of the same group also suggest that CA215 is expressed in membrane-bound forms as well as secreted forms [102]. Antibody digestion revealed that the recognition of CIgG by RP215 was significantly decreased when the N-linked glycosylation was removed by PNGase, while the removal of O-linked glycosylation by O Glycosidase had no influence. These results suggest that the epitope recognized by RP215 is attached to CIgG through N-linked glycosylation. Furthermore, it has been shown that the epitope is located on Asn162 in the CH1 domain, different from the known N-glycan located on Asn297 in the CH2 domain of physiological IgG (see Figure 6) [103]. In 2018, Tang et al. revealed that the oncogenic function of this abnormal N-glycan is mediated through interaction with the integrin α6β4 complex [103]. This interaction is known to result in the activation of FAK and Src pathways, thus driving tumorigenesis and metastasis of different cancers [104,105,106,107,108]. RP215 can inhibit this interaction by targeting the corresponding N-glycan epitope [103].

### 2.8. Perspectives

Although some characteristics and protumoral effects of CIgG have been investigated and it is clear that B cells are not the only cell type to express Igs, there are still many open questions. The promoting effects on tumor growth in vitro and in vivo have been shown by blocking CIgG expression [12,13,16,17,41,42]. Similarly, cancer cell migration and colony formation were diminished by blocking CIgG production in in vitro assays. However, these in vitro results could partly not be reproduced in vivo in the absence of immune cells [21]. The results of experiments with immunodeficient mice hint towards the necessity of immune cells, especially of T cells, to enable tumor cells to optimally benefit from CIgG. Though CIgG-expressing tumor cells clearly benefit from the auto- and paracrine effects, they are not the only targets of these Igs. Platelets have been shown to be activated when CIgG binds to FcγRIIa and initiates its phosphorylation [71]. The thereby activated platelets encage tumor cells and protect them from immune cell recognition and attack, as well as shear stress in the circulation [56,109,110]. As a result of the known direct inhibition of T cells and the activation of supporting platelets, it might be worth a try to investigate whether or not CIgG also has a direct influence on myeloid-derived suppressor cells. These cells, only discovered in 2007, have exclusively been observed in TME of tumors with immune regulatory activity [111]. They promote regulatory T cells and the differentiation of fibroblasts into cancer-associated fibroblasts. Myeloid-derived suppressor cells themselves can often mature into tumor-associated macrophages [112,113,114].

The CAR-T cell technology is a potent method to use membrane-bound CIgG as a point of attack to link tumor cells to designed T cells. Additionally, CD3 may boost the immune reaction of such T cells. Two different Fab parts allow bispecific CAR-T cells to interact with more than one target simultaneously. The tumor cells and “normal” T cells would be brought into close proximity of each other, thereby supporting immune recognition of the tumor cells by tumor-specific T cells and also CD3-positive NK cells.

Furthermore, if RP215 is confirmed as a robust reagent for CIgG detection, an application as tumor biomarker may be presumed. Therefore, the cross-reactivity to other molecules must be excluded meticulously. For diagnostic relevance of CIgG, the suitability (I) as a biomarker in the blood circulation and (II) as a prognostic marker when stained in cancer tissues by immunohistochemistry must be verified by larger studies.

In addition, there is no knowledge on the systemic effects of CIgG depletion from tumor patients’ blood. It is imaginable that with improved progress of patients’ conditions, CIgG could be eradicated from the circulation. Therefore, extracorporeal depletion of CIgG by antibodies like RP215 loaded onto columns can be envisioned.

Although the expression of CIgG in cancer cells could be established as a fact, the knowledge on the signaling pathways involved is still very limited. Due to the described effects on several immune mechanisms, it seems also wise to suggest the investigation of CIgG expression in the context of immunotherapies.

The observation that normal IgG can trigger oxaliplatin resistance further supports the necessity to analyze the signaling pathways used by cancer cells to take advantage of IgG molecules—either omnipresent normal IgG or CIgG [51]. Due to the limited knowledge, there may be other circumstances in which (C)IgG complicates a patient’s therapy by currently still unknown pitfalls.

## Figures and Tables

**Figure 1 ijms-22-11597-f001:**
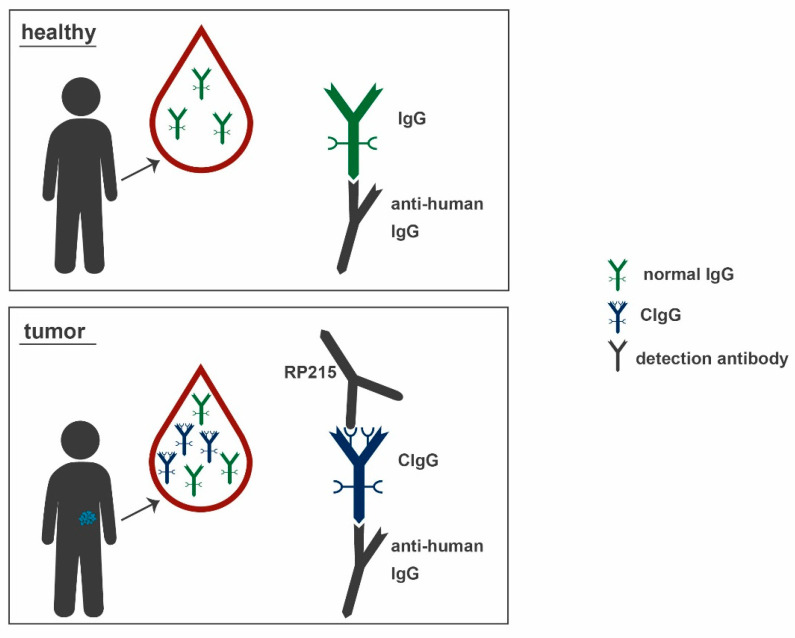
Normal human IgG (green) circulates in the bloodstream of healthy individuals and tumor patients, while CIgG (blue) is only present in the blood of tumor patients. Anti-human IgG antibodies (grey) detect both normal IgG and CIgG, but RP215 (also grey) specifically binds to the Asn162 epitope, solely present on CIgG. IgG: immunoglobulin, CIgG: cancer-derived IgG.

**Figure 2 ijms-22-11597-f002:**
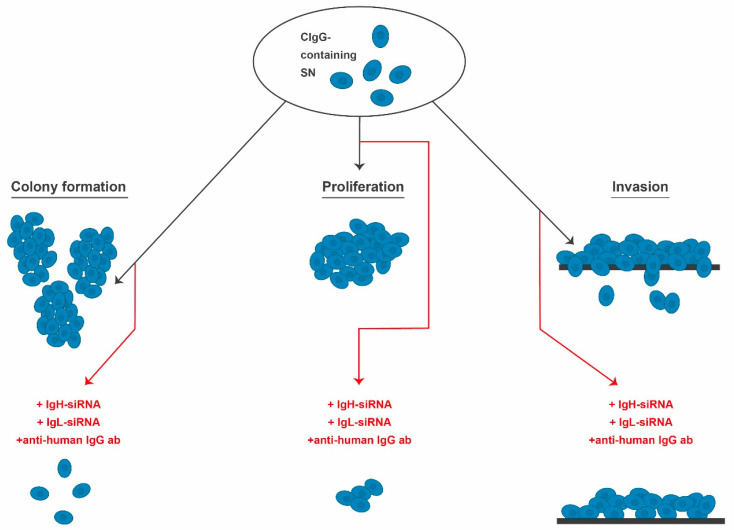
CIgG expression increases the cancer cells’ abilities in terms of colony formation, proliferation, and invasion. A reduction of these properties can be observed when targeting cancer cells with IgH-siRNA, IgL-siRNA, or anti-human IgG antibodies. CIgG: cancer-derived IgG, IgG: immunoglobulin G, ab: antibody, SN: supernatant, IgH: immunoglobulin heavy chain, IgL: immunoglobulin light chain, siRNA: small interfering RNA.

**Figure 3 ijms-22-11597-f003:**
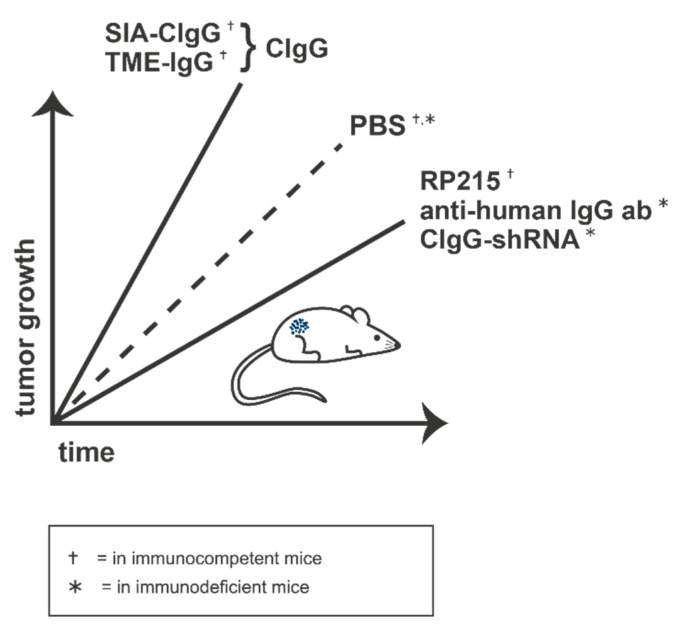
The protumoral effect of SIA-CIgG and TME-IgG causes an accelerated tumor outgrowth in vivo. Treatment with RP215, anti-human IgG antibody, or CIgG-shRNA, in contrast, reduces tumor growth rates in vivo. ab: antibody, PBS: phosphate buffered saline, CIgG: cancer-derived IgG, SIA-CIgG: sialylated CIgG, TME-IgG: IgG isolated from tumor microenvironment.

**Figure 4 ijms-22-11597-f004:**
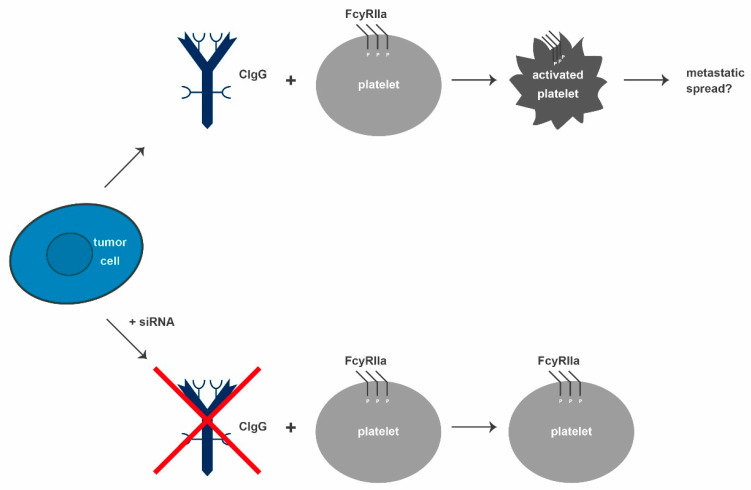
CIgG can bind to FcγRIIa on inactivated platelets and thereby increase phosphorylation levels. As a result, the platelets are activated and function as tumor progression facilitators in different ways. In addition, the metastasizing potential of tumors is reported to increase due to activated platelets. Blocking CIgG inhibits the CIgG-mediated platelet activation. CIgG: cancer-derived IgG, siRNA: small interfering RNA.

**Figure 5 ijms-22-11597-f005:**
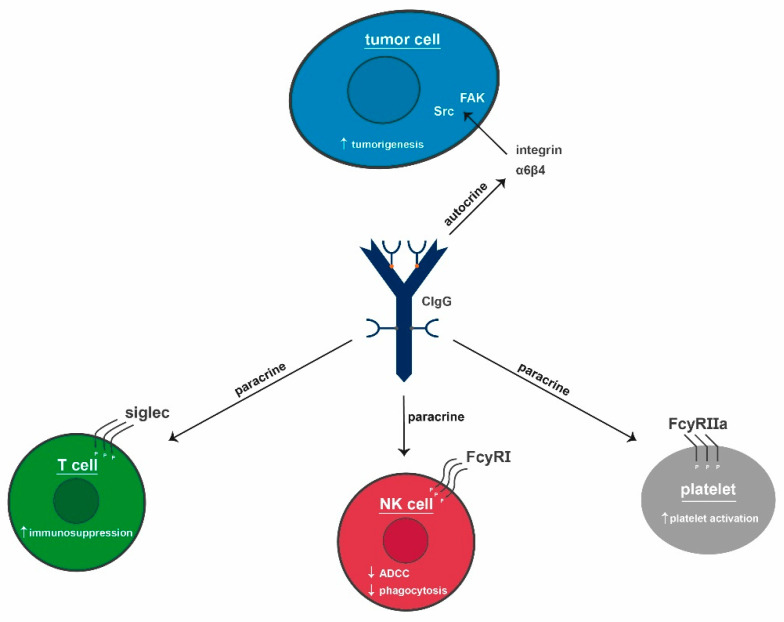
CIgG (blue) and its putative mode of action. CIgG is secreted by tumor cells and functions as an activator of integrin α6β4 on tumor cells (autocrine), siglecs on T cells, FcγRI on NK cells, and FcγRIIa on platelets (all paracrine). It drives tumorigenesis, immunosuppression, and platelet activation. Further, it impairs ADCC and phagocytosis. CIgG: cancer-derived IgG, NK cell: natural killer cell, ADCC: antibody-dependent cell-mediated cytotoxicity.

**Figure 6 ijms-22-11597-f006:**
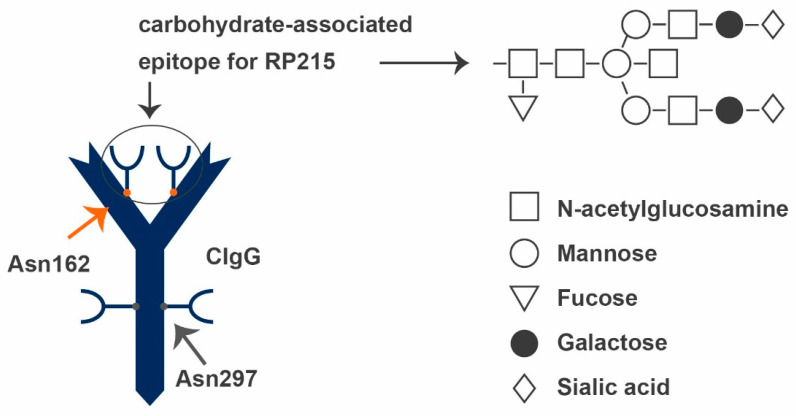
CIgG (blue) and its characteristic carbohydrate-associated epitope, Asn162 (red). Asn162 serves as an epitope for RP215 and can only be found at the CH1 domain of CIgGs. The epitope consists of N-acetylglucosamine, mannose, fucose, galactose, and N-linked sialic acid. The main part of Asn162 is consistent with Asn297. CIgG: cancer-derived IgG.

**Table 1 ijms-22-11597-t001:** Ig isotypes detected in tissues, cell lines and serum samples of different cancer entities.

Cancer Entity	Ig Isotype	Localization	Reference(s)
colon	IgG	tissue, cell line, serum	[12,14,17,32]
breast	IgG, IgA	tissue, cell line, serum	[12,15,32,33]
liver	IgG	tissue, cell line, serum	[12,32]
lung	IgG	tissue, cell line, serum	[12,32]
ovarian	IgG	tissue, cell line, serum	[12,32,34]
cervix	IgG, IgA, Igκ	tissue, cell line, serum	[12,32,35]
kidney	IgG	serum	[32]
lymphoma	IgG	serum	[32]
esophageal	IgG, Igκ, Igλ	tissue, cell line, serum	[36]
stomach	IgG	serum	[32]
pancreatic	IgG	tissue, cell line, serum	[32,37,38]
prostate	IgG, Igκ	tissue, cell line, serum	[32,35,39]
gastric	IgG	tissue, cell line	[40,41]
bladder	IgG	tissue, cell line	[13,42]
renal	IgG	tissue, cell line	[43,44]
nasopharyngeal	IgA, Igκ	cell line	[45,46]
laryngeal	IgM	tissue, cell line	[47,48]
oral	IgG, IgA	tissue, cell line	[49,50]
papillary thyroid	IgG	tissue	[18]
fibrous histiocytoma	IgG (serum only)	serum	[32]
malignant fibrous histiocytoma	IgG (serum only)	serum	[32]
fibroma	IgG (serum only)	serum	[32]
fibrosarcoma	IgG (serum only)	serum	[32]
leiomyoma	IgG (serum only)	serum	[32]
leiomyosarcoma	IgG (serum only)	serum	[32]
rhabdomyosarcoma	IgG (serum only)	serum	[32]

## Data Availability

Not applicable.

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
