# Peer review of "Cancer-Cell-Derived IgG and Its Potential Role in Tumor Development"

_ijms, 2021, doi:10.3390/ijms222111597_

Round 1

Reviewer 1 Report

The review by Said Kdimatioo et al. provides an overview of cancer cell-derived IgG and its potential role in tumor development, especially a specific carbohydrate-associated epitope of cancer-derived IgG recognized by RP215. It’s comprehensive and of great interest. However, the content of the review in some sections is sprawling, the following improvement need to be made:

  1. It seems conceptually wrong in the Abstract and Introduction section, such as IgG sequences are highly conserved, a variable region as Fab and the constant region as Fc. In fact, Fab comprise all variable regions and part of constant region of Ig heavy and light chain, while Fc, that is fragment crystallizable, comprises CH2 and CH3 domains of Ig heavy chain. Also, the sentence on line 42 seem to be not accurate, and it should be that different Ig gene segments are rearranged and assembled to form the variable region to bind different antigens.
  2. The logic of the review may need further improvement. For example, Ig repertoire in cancer cells and expression of tumor cell-derived IgG should be defined. More details about the sequence characteristics of CIgG derived from various tumor tissues should be discussed. Additionally, based on wide expression of Ig in different cancer cells, the Table should be advised to show the Ig expression profile in cancer tissues. Also, it has been described in effects of CIgG, and in vivo experiments should also be used to describe its function but not belong to a separate part. Besides, it is better to directly introduce how the CIgG activates platelets but not to mainly describe the association of CD62P in platelets activation.
  3. It seems to be inappropriate for Figure 1, that is described that Ig light chains in the cytoplasm of papillary thyroid cancer tissue. And it is confusing about Figure 5 that is about no significant differences were detectable in the immunodeficient mice.
  4. The IgG signaling is a critical part of this review, which should be organized logically. And the figure should be added to summarize the mechanism and function. In addition, it has been reported cancer-derived IgG activates integrin-FAK signaling in lung squamous cell carcinoma cells (Ref. Tang J., et al. Cancer Lett. 2018, 430, 148–159), so these results are suggested to add to the review.
  5. Too much details of referred results has been described, it should be better to be amplified. The labeling of subheadings is confusing, such as the missing 2.6, and one level down from IgG signaling consists of physiological and CIgG. It should beless ambiguous.
  6. Most sentences in the review area bit obscure, such as Line 75, 104, 128, 224, 250, etc. Some spelling error should also be corrected. In brief, the manuscript would benefit from English-language revision.

Author Response

Reviewer 1

The review by Said Kdimatioo et al. provides an overview of cancer cell-derived IgG and its potential role in tumor development, especially a specific carbohydrate-associated epitope of cancer-derived IgG recognized by RP215. It’s comprehensive and of great interest. However, the content of the review in some sections is sprawling, the following improvement need to be made:

  1. It seems conceptually wrong in the Abstract and Introduction section, such as IgG sequences are highly conserved, a variable region as Fab and the constant region as Fc. In fact, Fab comprise all variable regions and part of constant region of Ig heavy and light chain, while Fc, that is fragment crystallizable, comprises CH2 and CH3 domains of Ig heavy chain. Also, the sentence on line 42 seem to be not accurate, and it should be that different Ig gene segments are rearranged and assembled to form the variable region to bind different antigens.

Answer: We are thankful for the hint and corrected the introduction part. The definitions have been rewritten, so the statements are technically correct now. Based on this, we checked the rest of the definitions to avoid similar mistakes.

  1. The logic of the review may need further improvement. For example, Ig repertoire in cancer cells and expression of tumor cell-derived IgG should be defined. More details about the sequence characteristics of CIgG derived from various tumor tissues should be discussed. Additionally, based on wide expression of Ig in different cancer cells, the Table should be advised to show the Ig expression profile in cancer tissues. Also, it has been described in effects of CIgG, and in vivo experiments should also be used to describe its function but not belong to a separate part. Besides, it is better to directly introduce how the CIgG activates platelets but not to mainly describe the association of CD62P in platelets activation.

Answer: The Ig repertoire in different cancer entities has been collected and represented in table 1 with the corresponding citations. Thank you for the advice to clearly outline the Ig expression in cancers in the form of a table. The knowledge about the sequence characteristics of CIgGs is still limited. The main findings have been described and it has been stated that CIgGs shows a restricted pattern of rearrangement. Following your instruction, we combined the parts of in vitro and in vivo effects to improve the structure of the review. The chapter about platelet activation has been shortened to keep the main statement clear. The authors thank you for your guidance.

  1. It seems to be inappropriate for Figure 1, that is described that Ig light chains in the cytoplasm of papillary thyroid cancer tissue. And it is confusing about Figure 5 that is about no significant differences were detectable in the immunodeficient mice.

Answer: Figure 1 was indeed wrongly located and we apologize for this mistake. It has been corrected and the position has been changed. Also, figure 5 (figure 3 in the revised manuscript) has been improved.

  1. The IgG signaling is a critical part of this review, which should be organized logically. And the figure should be added to summarize the mechanism and function. In addition, it has been reported cancer-derived IgG activates integrin-FAK signaling in lung squamous cell carcinoma cells (Ref. Tang J., et al. Cancer Lett. 2018, 430, 148–159), so these results are suggested to add to the review.

Answer: According to this very helpful suggestion, a figure has been added to the review, summarizing the mentioned and described signaling receptors for CIgG. The integrin-FAK-Src pathway has been described and shown in the figure and the citation has also been added.

  1. Too much details of referred results has been described, it should be better to be amplified. The labeling of subheadings is confusing, such as the missing 2.6, and one level down from IgG signaling consists of physiological and CIgG. It should beless ambiguous.

Answer: The whole review was shortened and redundant content was deleted to clarify the main findings. Subheadings were changed and corrected. Again, we want to thank this reviewer for this mindful hint – the manuscript reads easier now!

  1. Most sentences in the review area bit obscure, such as Line 75, 104, 128, 224, 250, etc. Some spelling error should also be corrected. In brief, the manuscript would benefit from English-language revision.

Answer: Spelling errors as well as wrong or obscure sentences were corrected and the language of the whole review was checked again by the coauthor CSM, which is a native speaker of American English.

Reviewer 2 Report

“Cancer cell-derived IgG and its potential role in tumor development” by Said Kdimati et al.

This review focuses on the expression of immunoglobulin proteins in malignant cells and potential clinical and basic biological significance of this phenomenon. Although detection of IgG in cancerous cells has been identified and reported in several different labs, the importance and practical utilization of cancer-derived antibodies as biomarkers of therapeutic targets are not well understood. Therefore, a brief summary of this novel area of biomedical research is timely, well justified and important.

The review is well prepared and logically organized. All available information is included and discussed. The only minor problem is the absence of figure legends. They should be included and described figures in all details. There are only a few minor comments which might improve this interesting and important review.

Minor concerns:

  • First several sentences in Introduction require references
  • “(Fc) of an antibody forms a bond with other immune cells” – please correct to …specific receptors on other immune cells
  • 45-47: IgG is used for treatment not only immunodeficiencies but many other immune-mediated diseases. Please correct
  • In 2.1, the authors states (based on ref 9) that normal epithelial cells express IgH. This should be slightly expanded since it is unclear whether only malignant or also normal epithelial cells can produce immunoglobulins or their parts
  • Can the authors comment or briefly discuss unusual Cig in malignant plasma cells?

Author Response

Reviewer 2

This review focuses on the expression of immunoglobulin proteins in malignant cells and potential clinical and basic biological significance of this phenomenon. Although detection of IgG in cancerous cells has been identified and reported in several different labs, the importance and practical utilization of cancer-derived antibodies as biomarkers of therapeutic targets are not well understood. Therefore, a brief summary of this novel area of biomedical research is timely, well justified and important.

 The review is well prepared and logically organized. All available information is included and discussed. The only minor problem is the absence of figure legends. They should be included and described figures in all details. There are only a few minor comments which might improve this interesting and important review.

Answer: All figure legends were added to the manuscript; we apologize for this thoughtlessness and want to thank this reviewer for the hint.

 Minor concerns:

  • First several sentences in Introduction require references

Answer: Required references were added to the introduction accordingly.

  • “(Fc) of an antibody forms a bond with other immune cells” – please correct to …specific receptors on other immune cells

Answer: The sentence has been changed as suggested.

  • 45-47: IgG is used for treatment not only immunodeficiencies but many other immune-mediated diseases. Please correct

Answer: Literature research was done and the part of IVIG usage was extended accordingly.

  • In 2.1, the authors states (based on ref 9) that normal epithelial cells express IgH. This should be slightly expanded since it is unclear whether only malignant or also normal epithelial cells can produce immunoglobulins or their parts

Answer: To better explain the background of non-B cell-derived Ig, the known Ig expression was researched again and represented in the introduction part.

  • Can the authors comment or briefly discuss unusual Cig in malignant plasma cells?

Answer: We want to thank this reviewer for this mindful hint; Ig expression in malignant B cells has been described in the introduction. Plasma cell neoplasms as an origin of non-typical Igs were added to the manuscript.